# Detection of KPC, NDM and VIM-Producing Organisms Directly from Rectal Swabs by a Multiplex Lateral Flow Immunoassay

**DOI:** 10.3390/microorganisms9050942

**Published:** 2021-04-27

**Authors:** Alexandra Vasilakopoulou, Polyxeni Karakosta, Sophia Vourli, Eleni Kalogeropoulou, Spyros Pournaras

**Affiliations:** Clinical Microbiology Laboratory, Medical School, “Attikon” University General Hospital, National and Kapodistrian University of Athens, 1 Rimini Street, 12462 Athens, Greece; dralexandravasilakopoulou@gmail.com (A.V.); p_karakosta@hotmail.com (P.K.); svourli@med.uoa.gr (S.V.); pmargos@yahoo.gr (E.K.)

**Keywords:** carbapenemase detection, immunochromatography, fecal sample, carbapenem resistance, carbapenemase-producing organisms

## Abstract

We report a preliminary evaluation of the NG-Test CARBA 5 immunochromatographic assay for detecting carbapenemases directly from rectal swabs on the same day of sampling. Thirty fecal swabs were examined for carbapenemase-producing organisms (CPOs) by conventional culture, PCR, and NG-Test CARBA 5. Each sample was tested by the immunochromatographic assay five times, including direct testing and incubation in trypticase soy broth for 1, 2, 3, and 4 h. Twenty patients yielded CPOs by culture. Immunochromatographic and PCR results were concordant and detected the same 25 carbapenemases (11 KPC, 8 VIM, and 6 NDM). In five cases, we detected co-carriage of KPC and VIM. Compared with PCR, the sensitivity of NG-Test CARBA 5 for the detection of KPC, VIM, and NDM was 80% without incubation, 88% with one hour, 92% with two, and 100% with three hours incubation, while specificity was 100% for all time points. All samples containing adequate fecal content were detected by NG-Test CARBA 5 concordantly with PCR, without incubation. NG-Test CARBA 5 is a reliable test that rapidly detects the presence of carbapenemases at the same day of sampling, directly from rectal swabs. It thus provides early information to guide antimicrobial treatment and infection control interventions.

## 1. Introduction

The emergence and global dissemination of carbapenem-resistant Gram-negative bacteria (CRGN) limits therapeutic alternatives and represents a major public health threat. Infections by CRGN result in high morbidity and mortality while there is also a considerable impact in economy and society [1].The magnitude of this problem was assessed in a recent study that recorded the infections by antimicrobial-resistant bacteria and their outcomes in countries of the EU and the broader European Economic Area for 2015. Using the European Antimicrobial Resistance Surveillance Network data, this report calculated that there were approximately 670,000 infections with resistant bacteria, resulting inapproximately 33,000 attributable deaths and 870,000 disability-adjusted life-years (DALYs). Most importantly, it was noted that Greece and Italy had the highest burden among all participating countries and for Greece, most of the infections were due to carbapenem-or colistin-resistant bacteria [2].Hospitals and especially intensive care units seem to serve as the breeding grounds for multidrug resistant organisms [3]. Asymptomatic rectal colonization by these organisms usually precedes infection and constitutes a reservoir for transmission that needs to be identified rapidly and efficiently [4].The production of carbapenemases is the most prevalent mechanism of carbapenem resistance among CRGN [5]. Screening for intestinal carriage of carbapenemase-producing organisms (CPO) among hospital patients is very important, as it may guide the empirical treatment and support infection control and antimicrobial stewardship strategies [6]. Most CPOs harbor one or more of the following five most common carbapenemase families: KPC, NDM, VIM, OXA-48-like, and IMP [7].

According to a recent study, the CPO intestinal carriage rate among ward patients (excluding intensive care and other special units) in Attikon University Hospital, School of Medicine, National and Kapodistrian University of Athens, Athens, Greece was found to be quite high (8.2%) [8]. In that respect, the rapid and accurate identification of patients carrying CPOs is essential for our hospital both in terms of infection control as well as for the selection of the appropriate empirical therapy of patients.

Rapid genomic and phenotypic methods are currently available to accelerate the identification of carbapenemase-producing CRGN. Most of these methods are performed from colonies and thus require overnight incubation for the microorganisms to grow. Moreover, molecular methods that can be performed from colonies or directly from samples are relatively expensive and require specific instrumentation [6,9]. In order to overcome these problems, we attempted to implement a rapid phenotypic test directly in rectal samples.

NG-Test CARBA 5 is an in vitro rapid and visually read multiplex immunochromatographic assay that detects one or more of all the above five common types of carbapenemase enzymes (KPC, OXA-48-like, IMP, VIM, NDM) in a single strip. The test uses monoclonal antibodies that individually recognize each of the five carbapenemases. A control line has to be observed for the test to be considered valid.

In this study, we tested and report preliminary observations about the performance of the NG-Test CARBA 5 assay (NG Biotech, Guipry, France), for detecting these carbapenemases directly from rectal swabs on the same day of receiving the sample, in comparison with PCR and conventional culture. NG-Test CARBA-5 has already been evaluated for detecting CPOs directly from positive blood cultures with a sensitivity and specificity of 97.7% and 96.1%, respectively [10]. To the best of our knowledge this is the first study that tests the performance of NG-Test CARBA 5 directly in rectal samples, which would be very fast and convenient for the clinical laboratories.

## 2. Materials and Methods

The study took place at Attikon University Hospital, where CPOs are endemic. We tested samples of 30 patients hospitalized in our general ICU, which at the time of the study had 17 beds. All ICU patients are tested weekly for CRGN carriage; this study included 30 patients randomly selected among those screened regularly. The patients’ rectal swabs were screened for CRGN carriage using our laboratory’s routine surveillance conventional cultures, by NG-Test CARBA 5 (NG Biotech) on the first day of receiving the sample and by PCR, which served for comparison. The rectal samples tested were collected in Amies liquid medium (e-Swab, Copan, Brescia, Italy).

Each sample was tested by our standard of care method for the detection of CRGN, which includes culturing in MacConkey agar with 1 mg/L meropenem. Microorganisms were biochemically identified and tested for antibiotic susceptibility by the Phoenix 50 automated microbiology system (BD Diagnostic Systems, Sparks, MD, USA), according to EUCAST 2020 guidelines and breakpoints. The combined disk test was used for screening carbapenemase production using meropenem 10 μg disks (BIO-RAD, Marnes la-Coquette, France) with or without inhibitors [phenyl boronic acid (PBA) and ethylenediaminetetraacetic acid (EDTA)] [11]. The guidelines of the European Committee on Antimicrobial Susceptibility Testing (EUCAST) were applied for the detection of resistance mechanisms and specific resistances of clinical and/or epidemiological importance [12]. A meropenem disk with PBA and EDTA was also included to detect double carbapenemase producers (serine and metallo-β-lactamase, respectively) [11,13].

After plating in MacConkey agar, the residual rectal swabs were tested with an adaptation of the manufacturer’s protocol for NG-Test CARBA 5. Instead of using a colony, rectal swab samples in Amies medium were incubated at 37 °C in 6 mL of trypticase soy broth (TSB) in the presence of 0.25 mg/L meropenem. The test was performed five times for each patient by centrifuging 1 mL of the broth for 10 min with a relative centrifugal force of 1800× *g* (3000 rpm), starting at time zero (0 h, no incubation), and then after 1 h, 2 h, 3 h, and 4 h of incubation. The supernatant was discarded, and the pellet was resuspended with the lysis buffer provided in the NG-Test CARBA 5 kit (four drops, 120 μL) and processed following the manufacturer’s instructions. The presence of carbapenemases was confirmed by multiplex PCR [14] detecting *bla*_OXA-48_, *bla*_KPC_, *bla*_NDM_, *bla*_IMP_, and *bla*_VIM_. DNA extraction was performed from the broth (100 μL) after incubating for two hours (PureLink Genomic DNA Kits, Invitrogen, Life technologies, Carlsbad, CA, USA).

Since we tested only residual rectal swabs from the laboratory’s routine weekly surveillance culture program, there was no need for requesting patient consent.

## 3. Results

Among the 30 patient samples tested, we detected 20 cases positive for CRGN, which carried a total of 25 carbapenemase genes; 15 patients carried one and five patients two carbapenemases each, while10 samples were negative. The distribution of carbapenemases in the rectal samples was 6 KPC, 3 VIM, 6 NDM, and 5 VIM along with KPC. We did notidentify patients carrying OXA-48-like or IMP; IMP was never detected among *Klebsiella pneumoniae* in Greece and OXA-48 producers were rare in our hospital at the time of the study. The lateral flow immunoassay detected all the carbapenemases that were identified through molecular methods. The conventional phenotypic testing through culture yielded concordant results for 29 of the 30 samples.Inthe thirtieth case, the rectal culture resulted in the growth of *K. pneumoniae* producing KPC while the molecular diagnosis and the NG-Test CARBA 5 on the first day had detected the presence of both KPC and VIM (Table 1).

All the immunoassay results were valid, with the control line to be present. There were slight differences in the sensitivity of the immunochromatographic method for KPC and VIM detection, depending on the different incubation times of the broth. After three hours of incubation, the sensitivity and specificity for KPC, VIM, and NDM were 100% (no false positives, Table 2). All NDM and VIM carbapenemases were already detected from the first hour of incubation, while three and two KPCs were not detected in the first and second hour, respectively. Overall, 22/25 (88%) and 23/25 (92%) carbapenemases were positive in the first and second hour, respectively and 20/25 (80%) were detected without any incubation. All samples that contained visible fecal content (brownish color) gave concordant results with PCR, without incubation.

## 4. Discussion

In this preliminary report, we evaluated for the first time the NG-Test CARBA 5 applied directly in rectal swabs. All carbapenemases present were accurately detected on the same day of sampling, with sensitivity and specificity for KPC, NDM, and VIM to be 100% after a 3-h incubation in broth. Other evaluation studies of NG-Test CARBA 5 have similarly good performance results. In a study that took place in the Swiss National Reference Center for Emerging Antibiotic Resistance, in Fribourg, Switzerland, a collection of 73 carbapenem-resistant Gram-negative bacteria was tested. The collection included 45/73 Enterobacterales, 11/73 *Acinetobacter baumannii*, and 17/73 *Pseudomonas aeruginosa*. Isolates of the collection were resistant to carbapenems due to several different carbapenemases (OXA-48-types, KPC, NDM, VIM, IMP, OXA-23, OXA-40, OXA-58). Notably, four isolates co-produced two carbapenemases (NDM-1 and OXA-48-types). The NG-Test CARBA 5 allowed the identification of all OXA-48-types, KPC, NDM, VIM, and IMP carbapenemases even when there was co-carriage of two carbapenemases. As expected, the test did notdetect *A. baumannii* carbapenemases (OXA-23, OXA-40, OXA-58). Molecular identification of carbapenemases was in accordance with immunochromatography and the sensitivity and specificity of NG-Test Carba 5 were 100% [15]. In another study fromthe Antimicrobial Resistance and Healthcare Associated Infections Reference Unit, Public Health England, UK, 197 previously characterized bacterial isolates including 177 confirmed carbapenemase producers and 20 carbapenem-resistant but carbapenemase-negative isolates were studied and the overall sensitivity and specificity of the NG-Test CARBA 5 were 97.31%and 99.75%, respectively. All 14 isolates producing carbapenemases belonging to families other than the ‘big 5’ and the 20 carbapenem-resistant isolates without carbapenemases were correctly identified as negative with no false-positives [13].

It is promising that the performance on direct samples is comparable with that observed in previous evaluation studies of NG-Test CARBA 5 applied on cultured strains [13,15]. During our study, we did not have false positives nor false negatives. Furthermore, the test had an excellent performance in direct samples even when two different carbapenemases were present. The NG-Test CARBA 5 and PCR yielded totally concordant results. In one case, the conventional culture detected only *K. pneumoniae* producing KPC while immunochromatography and PCR detected KPC and VIM. Possible explanations for that could be the presence of a second microorganism carrying VIM in lower concentration than the KPC-producing *K. pneumoniae* that was not cultured in isolated colonies due to *K. pneumoniae* overgrowth or the presence of another *K. pneumoniae* strain with similar colony morphology with that of the KPC producer, which was not picked up for testing. Alternatively, a VIM-producing organism with low meropenem MIC (susceptible) and thus unable to grow on the 1 mg/L meropenem containing agar plate used might have carried the VIM gene.

The availability of a prompt result on the same day of sample collection can be very helpful for taking early infection control measures and reducing the spread of CPOs inside the hospital (6). Moreover, the immunochromatographic test is easy to perform, without requiring special machinery nor special skills. The result is visible within 15 min (and in many cases, it is readable even within five minutes). The protocol we followed for testing directly from rectal swabs is much quicker than the conventional culture method and lacks the high cost of the molecular methods. The fact that the test can quickly identify the intestinal carriage of different types of carbapenemases belonging to the five most common carbapenemase families is also important when choosing empirical therapy for severe infections or immunocompromised patients. Our results are comparable to those by Fauconnier et al. who tested the immunochromatographic assay OKN K-SeT test for rapid screening of CPO fecal carriage and reported 100% specificity and 96% sensitivity [14].

Limitations of the study are that we did notevaluate the test for IMP since it is a non-existent carbapenemase in Greece and, similarly, we did notdetect OXA-48-types because they are rare in our hospital and were notpresent among the relatively small numberofsamples that we included in the study. Furthermore, we did notperform sequencing of the carbapenemase genes detected nor molecular typing of the isolates cultured and thus we do notknow about the genetic diversity of the isolates and their resistance genes. However, the study used clinical samples from CPO carriers and not spiked, laboratory-generated, samples and it was performed in real-life hospital conditions. In addition, another limitation point is that this immunochromatographic test cannot detect newly emerging carbapenemases or rare ones compared to biochemical tests based on a color change such as the Rapidec CarbaNP or the β-carba test that can detect any carbapenemase activity [15].

Moreover, we have to stress that same-day testing requires a good-quality rectal swab sample in liquid Amies medium. We observed that poor samples (colorless with low turbidity—indicating low fecal content and presumably low microbial concentration) needed two to three hours of incubation in broth in order to detect carbapenemases while all the turbid (brownish) samples were positive without any incubation. It is thus plausible that samples containing adequate fecal content can be tested directly, saving time and workload, and that the test would be more effective if performed using fecal samples instead of rectal swabs.

This study has shown that the NG-Test CARBA 5 is a simple and inexpensive test, which is useful for the quick detection of gastrointestinal colonization by most of the carbapenemase producing CRGN. Another good feature of this method is that all of its reagents can be stored in room temperature for prolonged time.

In conclusion, the NG-Test CARBA 5 is a reliable immunochromatographic test that can detect rapidly the presence of carbapenemases from rectal swabs on the first day and thus provides important early information for infection control and antimicrobial stewardship. The rapidity and the easiness of the test make it particularly useful, especially in hospitals where CPOs are endemic.

## Figures and Tables

**Table 1 microorganisms-09-00942-t001:** Bacterial Culture, PCR, combined disc test with PBA and EDTA, and NG-Test CARBA 5 results for carbapenemase-positive rectal samples (Total N = 20).

No	Bacterial Isolate	PCR	Combined Disc Test with PBA and EDTA	NG-Test Carba 5
1	(1) *K. pneumoniae* (2) *P. aeuginosa*	*bla*_KPC_ + *bla*_VIM_	(1) carbapenemase inhibited by PBA (2) carbapenemase inhibited by EDTA	KPC + VIM
2	*P. aeruginosa*	*bla* _VIM_	carbapenemase inhibited by EDTA	VIM
3	*K. pneumoniae*	*bla* _KPC_	carbapenemase inhibited by PBA	KPC
4	*K. pneumoniae*	*bla*_KPC_ + *bla*_VIM_	carbapenemase inhibited by PBA + EDTA	KPC + VIM
5	(1) *K. pneumoniae* (2) No VIM producing organism isolated	*bla*_KPC_ + *bla*_VIM_	(1) carbapenemase inhibited by PBA	KPC + VIM
6	*K. pneumoniae*	*bla* _NDM_	carbapenemase inhibited by ΕDTA	NDM
7	*K. pneumoniae*	*bla* _KPC_	carbapenemase inhibited by PBA	KPC
8	*K. pneumoniae*	*bla* _NDM_	carbapenemase inhibited by EDTA	NDM
9	(1) *K. pneumoniae* (2) *P. aeruginosa*	*bla*_KPC_ + *bla*_VIM_	(1) carbapenemase inhibited by PBA (2) carbapenemase inhibited by EDTA	KPC + VIM
10	*K. pneumoniae*	*bla* _KPC_	carbapenemase inhibited by PBA	KPC
11	*P. aeruginosa*	*bla* _VIM_	carbapenemase inhibited by EDTA	VIM
12	*P. aeruginosa*	*bla* _VIM_	carbapenemase inhibited by EDTA	VIM
13	*K. pneumoniae*	*bla* _KPC_	carbapenemase inhibited by PBA	KPC
14	*K. pneumoniae*	*bla*_KPC_ + *bla*_VIM_	carbapenemase inhibited by PBA + EDTA	KPC + VIM
15	*K. pneumoniae*	*bla* _NDM_	carbapenemase inhibited by EDTA	NDM
16	*K. pneumoniae*	*bla* _NDM_	carbapenemase inhibited by EDTA	NDM
17	*K. pneumoniae*	*bla* _KPC_	carbapenemase inhibited by PBA	KPC
18	*K. pneumoniae*	*bla* _KPC_	carbapenemase inhibited by PBA	KPC
19	*K. pneumoniae*	*bla* _NDM_	carbapenemase inhibited by EDTA	NDM
20	*K. pneumoniae*	*bla* _NDM_	carbapenemase inhibited by EDTA	NDM

**Table 2 microorganisms-09-00942-t002:** Performance of the NG-Test CΑRΒΑ 5 directly from rectal swabs compared to PCR.

Carbapenemase	Broth Incubation Time (h)	TP	FN	Sensitivity	TN	FP	Specificity
KPC	0 h	7	4	63.6%	19	0	100%
	1 h	8	3	72.7%	19	0	100%
	2 h	9	2	81.8%	19	0	100%
	3 h	11	0	100%	19	0	100%
	4 h	11	0	100%	19	0	100%
VIM	0 h	7	1	87.5%	22	0	100%
	1 h	8	0	100%	22	0	100%
	2 h	8	0	100%	22	0	100%
	3 h	8	0	100%	22	0	100%
	4 h	8	0	100%	22	0	100%
NDM	0 h	6	0	100%	24	0	100%
	1 h	6	0	100%	24	0	100%
	2 h	6	0	100%	24	0	100%
	3 h	6	0	100%	24	0	100%
	4 h	6	0	100%	24	0	100%

## Data Availability

The data presented in this study are available on request from the corresponding author. The data are not publicly available due to patients’ privacy.

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
