# Peer review of "Detection of KPC, NDM and VIM-Producing Organisms Directly from Rectal Swabs by a Multiplex Lateral Flow Immunoassay"

_microorganisms, 2021, doi:10.3390/microorganisms9050942_

Round 1
Reviewer 1 Report
Vasilakopoulou et al. evaluated the efficiency and reliability of NG-Test CARBA 5 immunochromatographic assay for the detection of Carbapenem-resistant Gram-negative bacteria (CRGN) from rectal swabs. This assay could help early screening of the CRGN, especially in hospital environments when compared to laborious/time-consuming culturing and molecular biology techniques. The manuscript is written in a concise manner-easy to read and understand.
While this is a very preliminary report, current work is limited by the number of samples used for the study.
Minor comment:
Please check the placing of the reference number throughout the manuscript.
Author Response
Vasilakopoulou et al. evaluated the efficiency and reliability of NG-Test CARBA 5 immunochromatographic assay for the detection of Carbapenem-resistant Gram-negative bacteria (CRGN) from rectal swabs. This assay could help early screening of the CRGN, especially in hospital environments when compared to laborious/time-consuming culturing and molecular biology techniques. The manuscript is written in a concise manner-easy to read and understand.
While this is a very preliminary report, current work is limited by the number of samples used for the study.
Answer: Thank you for your kind comments. We have mentioned as a limitation of the study, the small number of samples (Line 191).
Minor comment:
Please check the placing of the reference number throughout the manuscript.
Answer: Thank you for pointing this out. We have corrected the placing of the reference number throughout the manuscript.
Reviewer 2 Report
Just for the Discussion
- The high percentage of positive cases for CRGN is surprisingly high. Can these organisms be tested also in hospital environment besides the swabs from patients?
- Were the "positive" patients different from "negative" patients by their feelings or other characteristics of the sickness?
Author Response
- The high percentage of positive cases for CRGN is surprisingly high. Can these organisms be tested also in hospital environment besides the swabs from patients?
Answer: We have tested samples from surfaces in the ICU and we have recovered some of the multidrug resistant microorganisms present in the rectal samples. After thorough cleaning and decontamination of the ICU repeated sampling from the same surfaces was negative. The paper focuses on the new rapid way to detect carbapenemases directly from the rectal samples and thus we did not include this information in the manuscript.
- Were the "positive" patients different from "negative" patients by their feelings or other characteristics of the sickness?
Answer: Although this study didn’t focus in examining the risk factors for colonization with multidrug resistant microorganisms, we observed that the colonized patients had been usually hospitalized for longer time than the negative patients. As above, we did not include this information in the manuscript.
Reviewer 3 Report
Antibiotic resistance is a growing global menace. In this manuscript, Vasilakopoulou et al. compare NG-Test CARBA 5 and the traditional PCR approach to detect carbapenemases using 30 rectal swabs samples. This work will be of broad interest to readers of microorganisms journal. However, some items need to be addressed before publication.
Page 1, line 33: It would be nice for authors to add “The European Antimicrobial Resistance Surveillance Network” before “EARS-Net”.
Page 1, In the text, reference numbers should be placed in square brackets [ ], and placed before the punctuation.
Page 2, line 91-92: The combination of PBA and EDTA can be used to detect the co-existence of serine-β-lactamase and MBL. However, how do authors know they are KPC and VIM?
Page 3, line 109-112: It is hard to follow the math here. It would be nice for authors to provide the complete profile of all 20 positive cases (patient number, bacterial isolate, results from PCR, disc diffusion assay, and CARBA5) in Table 1.
Page 4, line 116: The authors claim that they get the “concordant results” from the conventional phenotypic testing. It would be nice for the authors to explain where the number 29 is from.
Author Response
Antibiotic resistance is a growing global menace. In this manuscript, Vasilakopoulou et al. compare NG-Test CARBA 5 and the traditional PCR approach to detect carbapenemases using 30 rectal swabs samples. This work will be of broad interest to readers of Microorganisms journal. However, some items need to be addressed before publication.
Page 1, line 33: It would be nice for authors to add “The European Antimicrobial Resistance Surveillance Network” before “EARS-Net”.
Answer: We have followed your suggestion and we have added the full name of EARS-Net.
Page 1, In the text, reference numbers should be placed in square brackets [ ], and placed before the punctuation.
Answer: We have followed your instructions. Now all the reference numbers are in square brackets and are placed before the punctuation.
Page 2, line 91-92: The combination of PBA and EDTA can be used to detect the co-existence of serine-β-lactamase and MBL. However, how do authors know they are KPC and VIM?
Answer: You are right, we have corrected the text in Lines 93-94: ‘to detect double carbapenemase producers (serine and metallo-β-lactamase, respectively).’
Page 3, line 109-112: It is hard to follow the math here. It would be nice for authors to provide the complete profile of all 20 positive cases (patient number, bacterial isolate, results from PCR, disc diffusion assay, and CARBA5) in Table 1.
Answer: Table 1 has been changed and now includes all the information that you suggested. Also, to make it clearer to the readers, we have added in the text Lines 111-112: ‘fifteen patients carried one and five patients two carbapenemases each, while 10 samples were negative’.
Page 4, line 116: The authors claim that they get the “concordant results” from the conventional phenotypic testing. It would be nice for the authors to explain where the number 29 is from.
Answer: We tested 30 rectal samples. In the new Table 1, we describe in detail the findings for each one of the positive cases and it is shown that in one sample (sample number 5) there was a discrepancy between conventional culture as compared to PCR and immunochromatography from the rectal sample. All the other samples (29) produced the same results also by conventional culture, PCR and by the new testing protocol of immunochromatography directly from the swab. To make it clearer to the readers, we have written in Lines 118-122: ‘The conventional phenotypic testing through culture yielded concordant results for 29 of the 30 samples. In the thirtieth case, the rectal culture resulted in the growth of K. pneumoniae producing KPC while the molecular diagnosis and the NG-Test CARBA 5 on the first day had detected the presence of both KPC and VIM (Table 1).’
Round 2
Reviewer 3 Report
The authors have addressed my concerns. The revised manuscript looks good.